# Soybean Seed Coat Cracks and Green Seeds—Predisposing Conditions, Identification and Management

Ernane Miranda Lemes *[ID] and Hugo César Rodrigues Moreira Catão [ID]

Instituto de Ciências Agrárias (ICIAG), Universidade Federal de Uberlândia (UFU), Uberlândia 38410-337, Brazil; hugo.catao@ufu.br

*   Correspondence: ernanefito@gmail.com; Tel.: +55-(34)-99661-2808

**Abstract:** Seed coat cracking and green seeds threaten soybean crop production. Seed coat cracking results from a complex interplay of genetic factors, environmental stresses, and crop management practices. Green seeds, linked to water deficit, nutritional deficiencies, and environmental stresses, exhibit reduced quality and viability. The intricate relationships between seed coat integrity and seed permeability, influenced by the lignin content, porosity, and color, play a pivotal role in seed germination, storage potential, and resistance to field stresses. These issues reverberate through the soybean agricultural supply chain. Strategic interventions are crucial to address these abnormalities and ensure soybean productivity. Seed germination and vigor are reduced due to seed coat cracking and green seeds, undermining food security and necessitating additional resources for disease management. The occurrence and identification of green seeds and seeds with cracks in the seed coat were also reported by identifying the genes and QTLs (quantitative trait loci) associated with these characteristics. Herbicides, commonly used in weed management, may offer a strategic approach to mitigating seed coat cracking and green seed occurrence. Understanding the complex interactions between the genetics, environmental factors, and management practices influencing seed abnormalities is essential as global climate change intensifies. This review emphasizes the need for integrated strategies, balanced plant nutrition, and cohesive phytosanitary management to mainly alleviate seed coat cracking and greenish occurrences in soybeans and other plant species.

**Keywords:** *Glycine max*; seed tegument rupture; seed chlorophyll; water stress; crop management; herbicides





## 1. Introduction

Soybean (*Glycine max* L.) production encounters numerous challenges throughout its crop cycle. Notably, seed coat cracking and the presence of green seeds jeopardize the grain and seed quality, posing significant threats to agricultural productivity and sustainability. Seed coat cracking results from a complex interaction of genetic factors, crop management, and stresses, including mechanical damage during harvest, plant diseases, insect pests, and variations in temperature and moisture during preharvest plant stages.

Imperfections of the seed coat are a way for environmental factors and pathogens to affect the seed quality. Seed coat breakdown begins at the final $R_6$ soybean growth stage before physiological maturity [1]. Initial tissue breakdown occurs around the hilum. As the seed matures and dries out, the fissures lengthen and usually become perpendicular to the hilum, exposing the parenchyma cells [1].

The green seeds are commonly attributed to water deficit, nutritional deficiencies, diseases, and high temperatures during seed development [2]. Greenish seeds can cause problems during emergence in the field, making it challenging to establish suitable stands for different cultivars. Furthermore, many seeds are often discarded during sowing due to the low physiological quality of green seeds [3]. The amount of chlorophyll present in soybean seeds can be affected depending on the stage of maturation and drying, as well

as climatic conditions that can affect normal maturation in the field [3]. The chlorophyll of seeds dehydrated quickly is not degraded at the same rate as seeds dehydrated slowly, which have 1 µg chlorophyll semente$^{-1}$. When quickly dehydrated, the seeds remain green and contain up to 29 µg of chlorophyll seed$^{-1}$ [4].

Green and cracked seeds exhibit compromised physiological quality, a shorter shelf life, reduced water absorption resistance, and lower germination potential. The relationships between seed coat integrity and characteristics, including permeability, are influenced by the seed coat lignin content, porosity, and color. These are pivotal in determining the seed vigor, storage potential, and resistance to field stresses (vigor).

Storage challenges also arise from seed coat cracking and green seeds, particularly in the face of climate-related stresses. The increased susceptibility of damaged seeds to biotic stresses such as diseases may require additional resources and agrochemicals, potentially conflicting with the production costs and sustainable agricultural practices. Addressing these seed abnormalities requires strategic interventions to ensure the resilience of soybean agriculture.

Furthermore, herbicides are agrochemicals used in weed management in many crop areas; they also serve as a direct tool in crop production, particularly in soybean desiccation, to standardize crop plants and expedite grain harvest to improve soil usage and the sustainability of agriculture by ensuring faster production in the same area. Using herbicides to terminate plants and anticipate the crop harvest can constitute a strategy to reduce the occurrence of seed coat cracking and green seeds for many plant species.

Addressing these seed abnormalities becomes crucial as global climate change intensifies and environmental stresses become more unpredictable. This review explores the intricate interactions among genetics, environmental factors, and management practices contributing to these seed issues in the literature. Aspects related to seed coat cracking and greenish soybean seeds have been covered. We also map the genetic loci associated with these characteristics and the use of herbicides in soybean crop management that can help mitigate these problems.

By comprehending the underlying causes and consequences of seed coat cracking and green seeds, we can develop strategies to minimize their impact, ensuring the overall productivity of soybean farming while safeguarding global food security.

## 2. Seed Tegument

The soybean seed comprises an embryonic axis and two cotyledons covered by the seed coat. The seed coat is the outermost layer of the seed, originating from the integuments of the ovule [primin (testa) and secundine (tegmen)] [5]. The seed coat is a plant organ that maintains quality and preserves the integrity of internal seed parts, protecting the embryo and reserve tissues against mechanical, environmental, entomological, and microbiological damage [6].

The seed coat furthermore acts on (i) the interactions between the seed's internal tissues and the external environment, (ii) gas exchange, and (iii) seed germination by interfering with the control mechanisms of seed development and dormancy [7]. The seed coat also supports the metabolism of nutrient transfer from the mother plant to the embryo, which later provides support and nutrition for the developing embryo [8,9].

The basic structure of the seed coat comprises four layers: the cuticle, epidermis (palisade cells), hypodermis (osteosclerids), and parenchyma (parenchymal cells), respectively, from the surface to the interior of the seed [10] (Figure 1).

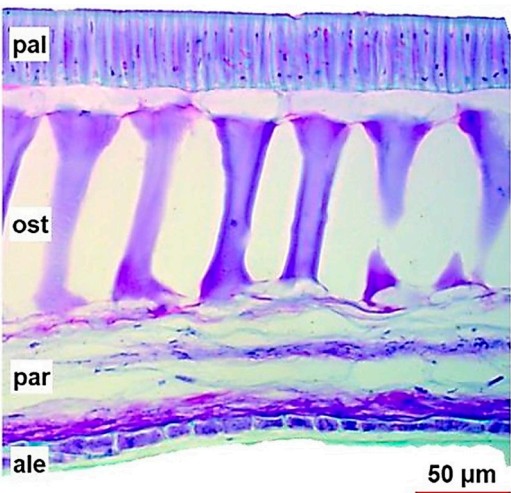

**Figure 1.** Anatomical section of the soybean seed coat close to the seed hilum. From external to internal coat layers: *pal*—palisade cells (epidermis), *ost*—osteosclerids (hourglass cells, hypodermis), *par*—parenchymal cells, and *ale*—aleurone layer. Source: [11].

The thickness of the soybean seed coat varies between 70 and 100 μm considering the four layers together, and there is variation among varieties (soybean cultivars). However, it should be noted that this physical characteristic is constant within each soybean variety and is genetically controlled [12]. The soybean seed coat consists mainly of insoluble carbohydrates, hemicellulose, cellulose, pectins, and phenylpropanoid polymers, that is, lignin [13].

Krzyzanowski et al. [14] reviewed the importance of lignin to soybean seed quality and highlighted that (i) high lignin contents in the soybean pod wall and seed coat are related to seeds with high physical, sanitary and physiological parameters; (ii) colored-coat soybean seeds (black or brown) have a higher seed quality due to their higher lignin content and the presence of anthocyanin; and (iii) boron presents an inverse relationship with lignin content.

The production and maintenance of seeds with intact seed coats must be a primary goal since an integral seed coat protects the embryo and contributes to the final quality of the soybean seed. The hardness of the seed coat is a hereditary characteristic controlled by one or a few genes of the maternal genotype that are highly influenced by the environment [15]. However, the environment can unlock non-genetic changes in the seed coat, such as thickness and composition, which do not persist beyond one generation [6].

Seed coat cracks and green seeds thus compromise soybean seeds' structural and physiological integrity; these vulnerabilities, particularly prominent in the late $R_6$ (full-size green beans at one of the four uppermost nodes) to early $R_7$ soybean phenological stages (pods yellowing, 50% of leaves yellow, physiological maturity), adversely affect the final quality and viability of the soybean seed, the germination potential, and overall seed performance in industry.

### 3. Seed Coat Cracking

A common disorder affecting soybean seed integrity is seed coat rupture. This condition is also termed seed tegument cracking and usually comprises elliptical, irregular (type I crack), or net-like cracks (type II crack) in the soybean seed coat [16]. The cracks can happen in one or many locations at any position and result from the separation of epidermal and hypodermal tissues from the inner layers of the seed coat [1,17].

Seed coat cracking usually happens for all the seeds in a soybean pod, and this tissue separation is regularly caused by the irregular contraction of the seed parenchyma or underlying tissues [1]. In recent years, the increasing release of soybean varieties driven by the high competitiveness of the market has also increased the incidence of

seed coat cracks [18]. This factor is still little explored in the field, and there are still uncertainties about the dimension and the actual influence of seed coat cracks on seed performance [19,20].

The seed coat crack starts in the late $R_6$ and early $R_7$ soybean phenological stages, where seeds achieve maximum weight just before physiological maturity [1,19,21]. It has also been reported that seed coat cracking occurs around the $R_6$ soybean phenological stage and is linked to the soybean variety. This seed damage exposes the cotyledonal tissues, negatively affecting germination and vigor.

The cracks on the seed also result from the mismatch between the growth of the seed cotyledons and coat [22]. The fast hydration of the seed cells during a period of water abundance (e.g., excessive rainfall) can exceed the seed tegument capacity, disrupting it. Such ruptures can occur during moisture loss, where the reduction in the seed coat water content is faster than the cotyledon reductions during the preharvest, causing the rupture of the outermost layers of the seed coat [1] (Figure 2).

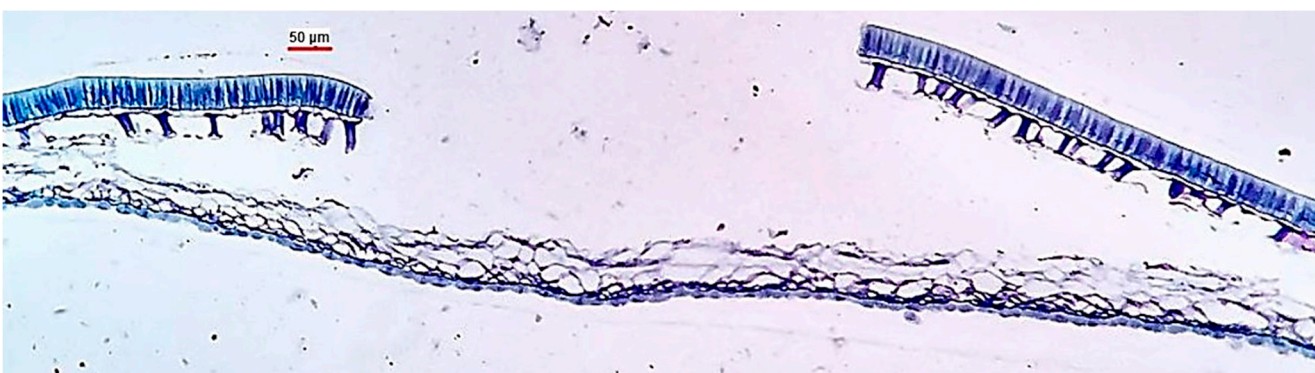

**Figure 2.** Anatomy of the soybean seed coat with a crack perpendicular to the seed hilum. Source: [11].

The soybean seed coat crack may be in the center of the cotyledon, flanking the seed hilum, or be minor cracks throughout the seed with no exact formation pattern. The different types of seed coat cracking are usually of genetic origin [23] or caused by many stresses such as mechanical damage, insect pests, diseases, and environmental variations such as dry heat during seed maturation and the alternation of rainy and dry periods in preharvest [17]. In addition, low temperatures after flowering can affect soybean coat integrity and may be associated with the accumulation of proanthocyanidins and subsequent lignin deposition [24].

Such differences in seed coat cracks indicate the need for studies with different varieties and the effects of this morphophysiological defect on seed quality and performance. Regardless of the origin, the seed coat crack severely affects soybean seeds' physical and physiological attributes.

The soybean seed longevity and tegument permeability are associated with the coat porosity, color, and integrity of the seed, and such permeability affects the seed vigor, storage potential, resistance to shrinking, imbibition damage, and the occurrence of seed pathogens [6,25–28]. According to Souza and Marcos-Filho [7], seed longevity correlates with the seed tegument permeability, and the lignin level in the seed tegument correlates with the seed's mechanical resistance. Capeleti et al. [29] and Menino et al. [30] also concluded that the seed coat lignin content and composition correlate positively with the resistance of soybean seeds to mechanical damage.

Ma et al. [31] observed that small cracks in the cuticle covering the palisade layer of soybean seed coats control the permeability of the seed to water. The authors concluded that the hardheadedness in soybeans could be partially attributed to a robust cuticular structure resistant to cracking and the cuticle of the palisade layer is the crucial factor determining the permeability property of a soybean seed coat.

Additionally, Lazarević et al. [32] reported that the micromorphological, morphological, and anatomical characteristics of the pea (*Pisum sativum* L.) seed tegument affect the damage extension caused by harvesting. Such a result observed for a plant species other than soybean indicates that care should be taken with the process and machinery involved with the grain harvest to reduce mechanical damage to the seeds. Non-calibrated machinery and/or harvest with the wrong seed water content could increase the occurrence of cracked and broken seeds.

Seed priming (increasing seed hydration followed by seed dehydration) could manage the damage in seed germination caused by seed coat cracking [33]. In the common bean, Mazibuko [34] reported that priming solutions with up to 50 mM of calcium salts [$CaCl_2$, $Ca(NO_3)_2$, $CaSO_4$] reduces the harmful effects of seed coat cracking. This effect is primarily due to improved seed cell wall integrity caused by available calcium. On the other hand, seed germination can be hastened by managing the cracks on the soybean coat. Chen et al. [35] used ultrasound to micro-crack the soybean seed coat to advance seed germination and metabolic performance of the sprout, indicating that seed pretreatment with ultrasound could use seed coat cracks to improve the initial response of the seed to the environment.

Adverse weather conditions during the grain filling and preharvest stages can contribute to the reduced quality of the seeds produced. As previously mentioned, extremes of temperature—either low or high—and rainfall—either the lack or abundance of rainfall—can impair the seed quality, especially after the seed has disconnected from the plant, accelerating their deterioration and reducing their seedling vigor when used as propagating material (seeds). After seed hydration begins the inevitable and irreversible process of seed deterioration, excessively hydrated seeds might present problems in storage, with diseases such as purple seed stain being caused by *Cercospora kikuchii* (Tak. Matsumoto and Tomoy.).

Shelar et al. [36] highlighted that a high seed moisture increases the flora of pathogens on soybean seeds, which plays an essential role in seed quality, deterioration, and the loss of viability during storage. Additionally, Machado et al. [37] reported that seed coat cracking in soybeans positively contributed to the moisture damage affecting the inner part of the seed. The authors also reported that *Fusarium* sp. and *Aspergillus* sp. were frequently observed in damaged seeds and seeds without cracks, but did not interfere with seed quality or serve as a gateway for the pathogens to cause damage during seed emergence.

Seeds with low physical and physiological quality, seeds that are malformed, cracked tegument and greenish, and unviable seeds are regularly present in any seed production field of soybeans or any other crop species. In order to counter climate-based stresses, especially high temperatures and excessive rainfall, little or nothing can be done to mitigate their deleterious effects. However, the magnitude of this damage to seeds can be minimized by, for example, crop nutrition management to ensure that no deficiencies or excess available plants nutrients occur (imbalanced plant nutrition regime can result in weak plants that are more susceptible to seed malformations), the prevention of diseases and insect pests (biotic factors can damage plant structures and photosynthesis and/or directly affect seed pods and the seed itself), adequate preharvest desiccation (correct timing of application and herbicide management can standardize seed maturation reducing seed exposition to field condition and seed problems occurrence), irrigation (lack of water during late seed development stages increase the occurrence of seed malformations—water must be provided according to crop needs), and the selection of soybean varieties less prone to such seed defects.

## 4. Green Seeds

As mentioned before, another common disorder that affects soybean seed integrity is the incidence of green seeds (greening seeds), which causes losses due to deterioration that is triggered by excess moisture in the seed. Water deficits and high temperatures at specific stages of seed formation can lead to the production of seeds that retain chlorophyll in their cotyledons, which leaves the seeds green [38,39]. França Neto et al. [2] also indicated

that high temperatures and the water deficit caused by drought during the grain filling and maturation phases could result in premature plant death and the intensification of green seed occurrence. According to Zorato [19], green seeds have a shorter longevity because wrinkles or dimples spread over the cotyledons, which makes seeds weak for water absorption and the beginning of the physiological processes of germination.

Soybean seeds may have a greenish color because of other factors. There are soybean varieties in which the seed chlorophyll is retained in the seed coat when the seeds are mature; this is a genetic trait [40]. Environmental biotic [diseases (e.g., root rotting, Asian soybean rust, purple seed stain) and insect pests (e.g., bedbugs)] and abiotic (weather extremes) stresses and the inadequate use of desiccants result in premature plant death and their forced maturation. Such forced maturation can cause greenish seeds to trigger germination, vigor, and viability reductions, preventing them from being stored for future use [41].

According to Ajala-Luccas et al. [42], (i) the soybean green seed anomaly seems to be more expressed as a function of thermal variation than water variation, (ii) there are commercial soybean varieties with a distinct sensitivity to the incidence of green seeds, indicating the potential for classical breeding, (iii) the seedling formation is significantly modified by the occurrence of green seeds, and (iv) the analysis of embryo protrusion can anticipate the diagnosis of a higher proportion of green seeds.

Moreover, Harbach et al. [43] summarizes anomalies such as green soybean syndrome, green stem disorder, and the greening effect in an exciting review. The authors discuss the terminologies for the symptoms observed and cited that biotic factors—like plant pathogens and insect pests—and abiotic factors—like herbicide and water stresses—interact and affect plant growth regulation and can cause each of the anomalies reviewed.

Chlorophyll degradation, the synthesis of soluble sugars, and modifications in soluble proteins occur during seed maturation. However, the chlorophyll of quickly dehydrated seeds does not degrade at the same rate as slowly dehydrated seeds (Figure 3). This occurs due to the very rapid translocation of reserves and lower photosynthesis rates [6]. Chlorophyll degradation is initiated during senescence by endogenous and external factors, such as water stress, reduced light, temperature variation, and increased ethylene content [2], as well as internal factors such as increased membrane permeability and pH changes.

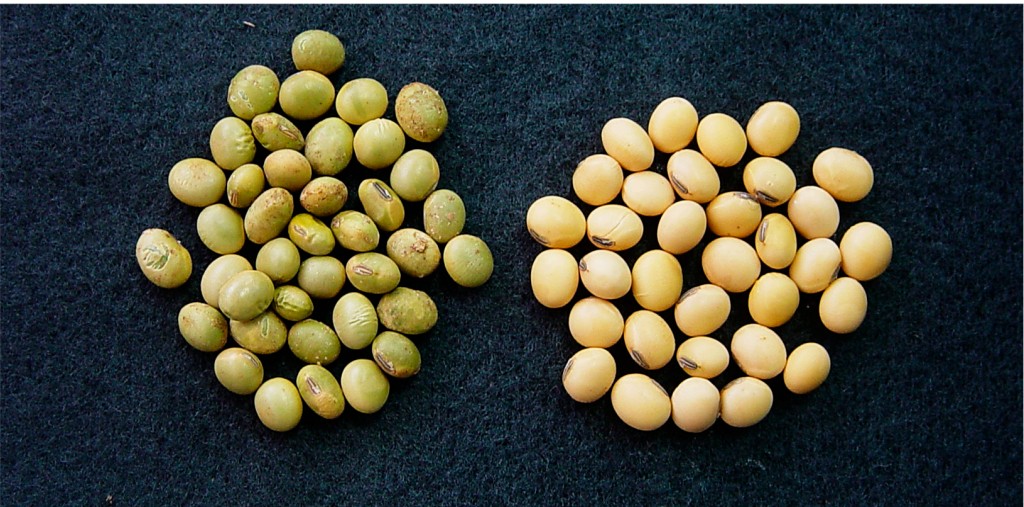

**Figure 3.** Quickly dried green soybean seeds (**left**) and yellow soybeans (**right**) slowly dried. Source: [2].

The enzymes chlorophyllase and magnesium chelatase decrease the chlorophyll in soybean seeds during maturation. The degradation of this pigment is an enzymatic process with an increase in chlorophyllides. However, this co-occurs with a non-enzymatic chemical process that increases pheophytins and other phytylated derivatives [44–46]. Porphyrin, a

heterocyclic chemical compound with a tetra-pyrrolic structure, is found in several natural organic molecules. It is a combination of chlorophyll a (bluish green) and chlorophyll b (yellowish green), which are always present in a 1:3 ratio (chlorophyll a: chlorophyll b) [47]. The only thing that separates them is that, in the side chain of chlorophyll a, the methyl radical (–CH$_3$) is replaced by an aldehyde group (–CHO) in chlorophyll b.

Pheophytins are formed by replacing chlorophyll's central magnesium (Mg) atom with hydrogen under acidic conditions [48]. The phytyl group is hydrolyzed by chlorophyllase to form chlorophyllide. Phaeophorbids, products of the loss of the phytyl group and Mg$^{2+}$, are brownish green and can undergo oxidative transformations that result in colorless degradation products [46]. In addition to stimulating carotenogenesis, which leads to the appearance of the yellow color, ethylene increases the activity of the enzymes oxidase and chlorophyllase [49,50], which are responsible for the degradation of chlorophyll and the disappearance of the green color [51].

França Neto et al. [2] suggests that increasing the temperature to more than 30 °C during the transfer of dry matter from the plant to the seed can cause severe damage to the production and quality of the seed. In this condition, reserves are quickly translocated, preventing the complete degradation of chlorophyll and promoting the production of green seeds of low quality and reduced performance [7]. Usually, the enzyme chlorophylase degrades chlorophylls, resulting in a normal soybean seed color. The activity of this enzyme is reduced in the final stages of maturation under climatic stress (hot and dry) conditions. These factors interfere with the normal process and cause chlorophyll degradation to accelerate or slow down [52]. During premature plant death and forced seed maturation, the activity of chlorophyllase is disrupted before chlorophyll is degraded.

Therefore, there is a need to update the information regarding the genes and QTLs associated with greenish seeds. A transcriptome analysis was performed to identify the changes in the gene expression in soybean seeds produced under non-stressed and combined heat and water stress conditions during the final stages of maturation [53]. Using an RT-qPCR approach, the authors demonstrated that combined heat and drought stresses affect the expression of genes in the chlorophyll degradation pathway and genes encoding chlorophyll-binding proteins. This impairment is caused by the expression of the *STAY-GREEN 1* and *STAY-GREEN 2* (*D1, D2*), *PHEOPHORBIDASE 2* (*PPH2*), and *NON-YELLOW COLORING 1* (*NYC1_1*) genes, which cause chlorophyll retention in the soybean seeds.

The genetic suppression of the green seed phenotype in soybeans could be an industrially helpful alternative. The *cytG* gene is cytoplasmically inherited from green soybean cotyledons and is maternally inherited [54]. This gene retains chlorophyll in leaves during senescence and in seeds during seed maturation, and is expected to stabilize seed quality by promoting chlorophyll degradation during soybean seed maturation.

The viability and vigor of the greenish soybean seeds deteriorated more, with this deterioration proportional to the percentage of seeds in which the cotyledons had green pigmentation, indicating that chlorophyll caused the physiological quality of the soybean seeds to decrease [55]. Tonello et al. [56] concluded in their review that soybean seeds should be harvested when they reach physiological maturity. However, harvesting is more usual when the seeds obtain a water content of around 14%. The authors also indicated that green seeds result from weather conditions, diseases and/or premature harvesting procedures, which impair the seed's physiological quality and negatively interfere with the batch quality and oil processing.

The quality of seed lots with 9% or more green seeds is significantly reduced to the point that their commercialization is not recommended [57]. The crop management methods used to prevent soybean green seeds and cracks are similar. As previously pointed out, appropriate management practices must occur to prevent soybean seeds from cracking and/or staying green (e.g., correction of plant nutritional imbalances, adequate control of diseases and insect pests, precise desiccation for harvest, and irrigation). In addition, selecting quality seed varieties that are resistant to crack and greening is essential to ensure a satisfactory harvest.

## 5. Mapping of QTLs for Seed Coat Cracking and Color Soybean Seeds

As mentioned in previous topics, the hardness of the soybean seed coat is a hereditary characteristic controlled by one or a few genes of the maternal genotype. Just as in greenish seeds, genes that encode binding proteins in the degradation of chlorophylls have been identified. These analyses are fundamental to understanding these effects on the quality of soybean seeds.

The magnitude of the effect, heritability of the trait, and interactions between genes depend on a study of the QTLs involved in soybean seeds with cracked coats and greenish seeds. To this end, the Soybase database (https://www.soybase.org/ accessed on 20 January 2024) was explored for information on QTLs that could identify the loci involved in seed coat cracks and colored soybean seeds. The main QTLs 'LOD Score' (LOD) are listed for each analyzed region of the genetic map and chromosomal regions for these characteristics (Table 1).

**Table 1.** Main QTLs 'LOD Score' (LOD) for each analyzed region of the genetic map and chromosomal regions for soybean seeds with cracked coats and greenish seeds.

| Seed Coat Cracking 1 | | | | |
|---|---|---|---|---|
| GWAS QTL Name | Original Name | LOD | Chromosome | Linkage Group |
| Seed coat cracking 1-g1 | qSC2-3 | 12.9 | Gm02 | D1b |
| Seed coat cracking 1-g2 | qSC6 | 6.8 | Gm06 | C2 |
| Seed coat cracking 1-g3 | qSC10-2 | 10.7 | Gm10 | O |
| Seed coat cracking 1-g4 | qSC12 | 5.1 | Gm12 | H |
| Seed coat cracking 1-g5 | qSC19-2 | 4.1 | Gm19 | L |
| Seed coat cracking 1-g6 | qSC20 | 6.3 | Gm20 | I |

| Seed Coat Cracking 2 | | | | |
|---|---|---|---|---|
| GWAS QTL name | Original name | LOD | Chromosome | Linkage Group |
| Seed coat cracking 2-g1 | qSCD15 | 5 | Gm15 | E |
| Seed coat cracking 2-g2 | qSCD20 | 6.1 | Gm20 | I |

| Seed Coat Color 1 | | | | |
|---|---|---|---|---|
| GWAS QTL name | Original name | Associated locus | *p*-value | Chromosome |
| Seed coat color 1-g1.1 | Seed coat color | Chr01_3568116 | $1.12 \times 10^{-7}$ | Gm01 |
| Seed coat color 1-g1.2 | Seed coat color | Chr01_3570619 | $1.12 \times 10^{-7}$ | Gm01 |
| Seed coat color 1-g1.3 | Seed coat color | Chr01_3576803 | $1.12 \times 10^{-7}$ | Gm01 |
| Seed coat color 1-g2 | Seed coat color | Chr01_53229579 | $4.17 \times 10^{-12}$ | Gm01 |
| Seed coat color 1-g3.1 | Seed coat color | Chr08_7800853 | $2.63 \times 10^{-35}$ | Gm08 |
| Seed coat color 1-g3.2 | Seed coat color | Chr08_8607374 | $2.63 \times 10^{-35}$ | Gm08 |
| Seed coat color 1-g3.3 | Seed coat color | Chr08_9079037 | $2.63 \times 10^{-35}$ | Gm08 |

| Seed Coat Color 3 | | | | |
|---|---|---|---|---|
| GWAS QTL name | Original name | Associated locus | R2 | Chromosome |
| Seed coat color 3-g1 | Black | ss715602762 | 0.62 | Gm08 |
| Seed coat color 3-g2 | Green | ss715580343 | 0.31 | Gm01 |
| Seed coat color 3-g3 | Yellow | ss715620481 | 0.41 | Gm15 |
| Seed coat color 3-g4 | Yellow | ss715580343 | 0.58 | Gm01 |
| Seed coat color 3-g5 | Yellow | ss715602763 | 0.56 | Gm08 |
| Seed coat color 3-g6 | Yellow | ss715593807 | 0.55 | Gm06 |

| Seed Coat Color 4 | | | | |
|---|---|---|---|---|
| GWAS QTL name | Original name | Associated locus | *p*-value | Chromosome |
| Seed coat color 4-g1 | Gm08:8236728 | Gm08:8241052 | $1.20 \times 10^{17}$ | Gm08 |
| Seed coat color 4-g2 | Gm08:8439795 | Gm08:8422490 | $1.31 \times 10^{15}$ | Gm08 |
| Seed coat color 4-g3 | Gm08:20745554 | Gm08:20702756 | $1.86 \times 10^{9}$ | Gm08 |

**Table 1.** *Cont.*

| Seed Coat Color 5 | | | | |
| --- | --- | --- | --- | --- |
| GWAS QTL name | Original name | LOD | Chromosome | Linkage Group |
| Seed coat color 5-g1 | qSC08 | 14.251 | Gm08 | D1b |
| Seed coat color 5-g2 | qSC11 | 8.112 | Gm11 | F |

The study of the loci controlling the quantitative traits (QTLs) associated with using the genetic maps obtained by molecular markers has allowed the genetic mapping of many traits of agronomic interest. With the advent of selection assisted by specific molecular markers for seed coat cracking and green soybean seeds, it may be possible to select superior genotypes quickly. Thus, the strategy that associates molecular markers with gene loci is essential for agriculture, and gene cloning is a valuable tool in soybean genetic improvement programs and the attainment of high-quality seeds.

## 6. Crop Desiccation as a Crop Management Tool

The herbicide active ingredient, dose, and soybean phenological stage for adequate crop desiccation are variable in the literature, but the results are mostly positive. The following results reported by many studies will indicate situations in which preharvest desiccation affected, or did not affect, the seed performance after harvest, and directly or indirectly indicate where seed coat cracks and green seeds affect the seed quality of soybeans and other crop species.

Daltro et al. [58], for example, observed that the physiological quality of soybean seeds were not affected by preharvest desiccation with paraquat (300 g ha$^{-1}$ of active ingredient), diquat (300 g ha$^{-1}$ of active ingredient), paraquat + diquat (150 + 150 g ha$^{-1}$ of active ingredient) or paraquat + diuron (300 + 150 g ha$^{-1}$ of active ingredient) in the R$_{6.5}$ (fully formed seeds, 50% yellow and 50% green pods) or R$_7$ soybean phenological stage. On the other hand, the authors reported that preharvest desiccation with glyphosate (1080 g ha$^{-1}$ of active ingredient) negatively affected the soybean seedlings' root system development and performance.

Sant'Anna Jr [59] reported that paraquat (400 g ha$^{-1}$ of active ingredient) anticipated soybean harvest faster than other herbicides and preserved more of the seed's physiological quality (soybean seed germination and vigor were not affected by paraquat). Ammonium glufosinate (400 g ha$^{-1}$ of active ingredient) and glyphosate (960 g ha$^{-1}$ of active ingredient) were less able to anticipate the soybean harvest and negatively affected the soybean seed physiological quality. Thus, ammonium glufosinate and glyphosate were not recommended as safe in the soybean seed production fields.

Instead, Bezerra [60] found that the desiccation of glyphosate-resistant soybeans [*Roundup Ready*® (RR) transgenic soybeans] with paraquat (400 g ha$^{-1}$ of active ingredient) at the R$_{5.7}$ (pods with seeds above 85% filling in at least one of the four last nodes of the main stem), R$_{6.5}$ (pods with completely developed seeds in at least one of the last four nodes of the main stem), and R$_{7.3}$ (one mature color pod in any central stem node) soybean physiological stages was not prejudicial to seed germination. The authors also concluded that paraquat desiccation at the R$_{7.3}$ soybean physiological stage was not prejudicial to soybean yield and seed vigor.

Lamego et al. [61] concluded that paraquat (400 g ha$^{-1}$ of active ingredient) desiccation at the R$_{7.3}$ (plants with more than 76% leaves and pods yellow) soybean physiological stage was not prejudicial to grain yield. The authors also concluded that the R$_6$ (pods with 100% filling and green leaves) and R$_{7.1}$ (onset to 50% of yellowing leaves and green beans) soybean physiological stages generated a faster seed germination speed. According to Finoto et al. [62], diquat (400 g ha$^{-1}$ of active ingredient) desiccation at the R$_{7.1}$ (onset to 50% of yellowing leaves and green beans) soybean physiological stage resulted in the best period, presenting a high germination rate and seed vigor. Santos et al. [63] also observed an anticipation period of 20 days in the soybean harvest when applying paraquat

(400 g ha$^{-1}$ of active ingredient) at the R$_7$ soybean physiological stage without losing a seed weight of one thousand and grain yield.

The application of preharvest desiccation with glyphosate can negatively affect the physiological quality of the soybean seeds and the seedling performance [64] depending on the soybean variety [65], and reduces the yield components even for glyphosate-resistant soybeans if glyphosate is applied in the vegetative phenological stages [66]. Albrecht et al. [67] found that the application of glyphosate (1440 to 2880 g ha$^{-1}$ of active ingredient) to RR soybeans in V$_6$ (plants with seven nodes and six trifoliates fully unfolded) and R$_2$ (open flower at one of the two uppermost nodes on the main stem) negatively affected seed germination and vigor, the seed performance in the tetrazolium test and the seedling biometrics.

However, Castro et al. [68] found that glyphosate (up to 4800 g ha$^{-1}$ of active ingredient) had no negative effect on seed quality or the enzymatic expression of glyphosate-resistant soybeans. Still, preharvest desiccation with glyphosate is highly discouraged, especially for the seed production of non-glyphosate-resistant soybeans, since it can cause reduced seed germination and possibly vigor problems.

Additionally, even glyphosate drift can alter the plant mineral composition. Cakmak et al. [69] found increased leaf concentrations of nitrogen, potassium, zinc, and copper in the vegetative stage of non-glyphosate-resistant soybean plants managed with glyphosate-simulated drift (low glyphosate doses $\leq 16.8$ g ha$^{-1}$ of active ingredient); however, the leaf concentration of calcium, magnesium, iron, and manganese generally decreased. The authors suggested that glyphosate may interfere with the uptake and retranslocation of the decreased nutrients, most probably by binding and thus immobilizing them. Such reductions would damage the soybean seed quality.

Albrecht et al. [70] also observed no differences between the ammonium glufosinate (400 g ha$^{-1}$ of active ingredient) formulations in the preharvest desiccation for defoliation, soybean pod maturation, and yield when desiccation was applied at the R$_{7.2}$ (about 68% of average physiological maturity) soybean phenological stage. However, the preharvest desiccation of soybean with ammonium glufosinate (494 g ha$^{-1}$ of active ingredient) reduced grain productivity. It increased the occurrence of green seeds compared to paraquat (570 g ha$^{-1}$ of active ingredient) and diquat (413 g ha$^{-1}$ of active ingredient) herbicides [71]. The authors also observed that preharvest herbicides had no effect on seed germination but indicated that the soybean phenological stage and climatic conditions could have affected the observed results.

Botelho et al. [72] found that preharvest desiccation negatively affected the soybean seed quality after 6 months of storage, especially preharvest desiccation with ammonium glufosinate (600 g ha$^{-1}$ of active ingredient). Reduced seed germination, seed vigor, and the lower mobilization of soluble protein and sugar in soybean seeds from plants desiccated with ammonium glufosinate (400 g ha$^{-1}$ of active ingredient) were previously reported by Delgado et al. [73].

According to Pereira et al. [74], soybean preharvest desiccation is able to anticipate soybean harvest; however, the success of the preharvest desiccation depends on the soybean variety, phenological timing of the herbicide application, herbicide active ingredient, and climatic conditions before and after desiccation. Additionally, Silva et al. [75] concluded that seed vigor could depend on soil management systems (tillage and no-tillage soil management systems). The authors also reported that soybean seed longevity was superior in the no-tillage system, but that preharvest desiccation reduces seed longevity.

In crops other than soybean, preharvest desiccation is intended for the same objectives: standardizing crop plants, expediting grain harvest, and improving soil usage by producing faster in the same area. In the common bean (*Phaseolus vulgaris* L.), the same plant family (Fabaceae) as soybean, preharvest desiccation with glufosinate (380 g ha$^{-1}$ of active ingredient), saflufenacil (73.5 g ha$^{-1}$ of active ingredient), or diquat (350 g ha$^{-1}$ of active ingredient) affected the integrity of the seed cell membrane, reduced the hectoliter weight and caused a loss in the sanitary quality [76]. However, the preharvest desiccation

of common bean with paraquat (400 g ha$^{-1}$ of active ingredient), ammonium glufosinate (400 g ha$^{-1}$ of active ingredient), and the mixture of paraquat + diuron (200 + 400 g ha$^{-1}$ of active ingredient), when 90% of the pods were brownish, caused no damage to the physiological quality of the seeds [77]. In addition, the authors observed a 12-day reduction in the common bean harvest period.

Coelho et al. [78] reported that the preharvest desiccation of common bean with paraquat (400 g ha$^{-1}$ of active ingredient) at 26, 30, or 34 days after flowering could anticipate the seed harvest by 16 to 24 days. The authors also found that preharvest desiccation had no negative effects on the common bean yield, seed germination, and vigor. Franco et al. [79] reported that the preharvest desiccation of common bean with diquat (400 g ha$^{-1}$ of active ingredient) could anticipate the seed harvest by 14 days. No prejudice to seed germination was observed for desiccations at and 83 days after sown (DAS); however, the authors detected reductions in seed production when desiccation happened at 83 DAS but not when desiccation occurred at and after 89 DAS.

The herbicide active ingredient can affect the compositional quality and physiological performance of seeds during propagation, reducing the seed germination rate and field vigor. In common beans, for example, Penckowski et al. [80] found that ammonium glufosinate (120 to 480 g ha$^{-1}$ of active ingredient) and ethephon (100 to 400 g ha$^{-1}$ of active ingredient), diquat (300 to 600 g ha$^{-1}$ of active ingredient), and ammonium glufosinate (300 g ha$^{-1}$ of active ingredient) applied as crop desiccants did not affect the common bean germination rate and field vigor; however, paraquat and glyphosate negatively affected the common bean germination rate, and glyphosate also reduced the common bean seed vigor.

In wheat (*Triticum aestivum* L.), preharvest desiccation can be managed to achieve the maximum harvest and physiological and sanitary results, since preharvest desiccation generates the uniform drying of the wheat panicle [71]. Cechinel [81] also observed that ammonium glufosinate (200 and 400 g ha$^{-1}$ of active ingredient) caused uniform wheat plant drying. Additionally, Lunkes [82] concluded that ammonium glufosinate (340 g ha$^{-1}$ of active ingredient) caused less damage to the wheat seed quality than glyphosate (720 g ha$^{-1}$ of active ingredient); this result indicates that some harm to the seeds can be expected when preharvest desiccation is applied to the wheat crop.

Determining the seed water content is also essential in identifying the potential damage caused by the preharvest desiccation performed at unappropriated timing, especially regarding the seed water content. Kehl [83] found that wheat preharvest desiccation at 30% water content reduced seed weight, and that wheat preharvest desiccation at 40% reduced seed vigor (depending on the herbicide active ingredient).

In canola (*Brassica napus* L.), Darwent et al. [84] observed that glyphosate (450 to 1700 g ha$^{-1}$ of active ingredient) applied to plants with a wide seed moisture range had little or no effect on the canola seed yield, weight, and germination, green seed occurrence, and oil content, except when glyphosate was applied to green pods and seeds with a high seed moisture content. The timing of herbicide application in preharvest is essential to achieve the best desiccation results. This timing is usually determined by factors such as the phenological stage of the crop, the degree of seed moisture, and the weather conditions in the days preceding and after desiccation.

The biggest problems occur when preharvest desiccation is anticipated outside the ideal period when the seeds still present a high water content; this will lead to losses of production and quality (compositional and physiological) and a high presence of herbicide residues. Late harvesting, with or without desiccation, increases production losses via the natural threshing of seeds and threshing during mechanical harvesting, favors the appearance of cracks in the seed tegument, and exposes them for a longer time to the attack of diseases and predatory insects. Therefore, the moment of herbicide application for preharvest desiccation is essential for this practice in order to generate the best results in seed production and quality without compromising the yield.

Plant desiccation is a viable alternative able to anticipate the soybean harvest. Still, according to Pereira et al. [74], it also depends on the soybean variety, the time of crop plant

desiccation, the active ingredient of the desiccant herbicide, and the lack of preharvest rainfalls. Here, it is imperative to mention that preharvest desiccation and any other spray application to field crops must be performed under favorable conditions. Always consider the herbicide's label recommendation regarding the adequate wind speed, temperature range, and air humidity to apply the herbicides and avoid losses in the efficiency of the desiccation.

## 7. Perspectives and Future Research

Future research endeavors should focus on refining the understanding of factors influencing the seed quality (e.g., the occurrence of soybean seed cracks and greenish) and crop performance on a long-term basis. Studies across diverse geographical regions should also be implemented to account for regional variations and the interactions of predisposing factors in each area.

Comprehensive knowledge of the intricate relationships among plant nutrition, phenological stages, environmental factors, stress intensities, and seed genetic characteristics is crucial for sustainable and resilient soybean production systems. Thus, the occurrence of soybean seed cracks and greening do not originate from one predisposing factor. A set of factors could interact to produce malformed seeds, and each factor's influence must be fully understood.

Studies could elucidate the interactions between herbicide types, dosages, soybean varieties, and phenological stages to establish more precise guidelines for effective preharvest desiccation. Such interactions need further exploration to optimize the balance between early harvest benefits and potential drawbacks. Assessing the influence of climatic conditions on preharvest desiccation outcomes could enhance the applicability of findings across diverse environmental contexts. Furthermore, exploring alternative desiccation methods and their implications for seed quality would be valuable.

The impact of preharvest desiccation on seed longevity and storage stability, as suggested by Silva et al. [75], warrants deeper investigation, including an examination of soil management systems. Beyond soybeans, similar research avenues could be pursued in other crops where preharvest desiccation practices may differ but share common concerns regarding seed quality, grain yield, and the influence of the timing of herbicide application.

Investigating the factors affecting seed quality should delve into the effects of specific microbial interactions on seed quality, exploring the possibility of beneficial or harmful interactions. Exploring the genetic basis of susceptibility or resistance to greenish discoloration and cracked seed coats in soybeans is a promising strategy to identify and incorporate the specific genes associated with these issues.

In-field monitoring (sensing technologies) and the integration of precision agriculture techniques (data-driven decision-making and variable-rate technologies) should also be explored to mitigate the occurrence and severity of damage to these seeds. The occurrence of natural field non-uniformities areas must be mapped, and localized soil problems must be corrected to reduce the incidence of seed abnormalities that reduce the seed lot quality in soybeans and other crop species.

## 8. Conclusions

Considering what has been observed in the literature, it is possible to infer that seed abnormalities, such as seed coat crack and greenish seeds, are a significant function of genetics, plant nutrition, climatic conditions, desiccation management, and the interaction among these factors. In short, seeds with cracks and/or greening are commonly observed when high temperatures associated with water stress occur during grain filling and maturation.

Another aspect of these seed anomalies is that the observed results are inconsistent across different crops and crop varieties. The factors affecting soybean crack occurrence differ from those affecting other plant species; this indicates the complexity of the factors involved in the occurrence of these abnormalities. Thus, nutritional deficiencies (or excesses),

climate variations, plant diseases, and insect pests may contribute to the occurrence of these problems in seeds. However, some occurrences of seeds with cracks and/or greening may also be associated with inadequate preharvest desiccation.

No specific herbicide was indicated to predispose crops to these seed abnormalities. It was recommended that glyphosate be avoided in the preharvest desiccation of non-RR soybeans, as it could further affect seed germination and vigor. The early application of desiccants, when the seed still has a high water content, significantly increases the occurrence of seeds with cracks and greening. Adequately choosing the timing of preharvest desiccation is essential to reduce these seed problems. In the case of soybean crops, the preharvest desiccation should happen in the late $R_6$ to early $R_7$ soybean phenological stage.

This review conveys that seed abnormalities will occur primarily in stressful conditions during flowering, seed formation, filling, maturation, and harvest. Cultivating soybean varieties resistant to these seed disorders, equilibrated plant nutrition, adequate preharvest desiccation, a good-quality harvest, and integrated disease and insect pest management can alleviate the occurrence of cracks and greening in soybean seeds. The available literature has not indicated anything other than these recommendations.

**Author Contributions:** E.M.L. and H.C.R.M.C.: conceptualization; writing of the original manuscript; review and editing; supervision. All authors have read and agreed to the published version of the manuscript.

**Funding:** This research received no external funding.

**Institutional Review Board Statement:** Not applicable.

**Informed Consent Statement:** Not applicable.

**Data Availability Statement:** Data are contained within the article.

**Acknowledgments:** The authors thank Daniel L. M. Machado, who contributed to the publication of this review; José de Barros F. Neto and Sheila B. Teixeira, who provided the images used; and the Instituto de Ciências Agrária (ICIAG) of Universidade Federal de Uberlândia (UFU) for its personnel and structural support. They also thank all previous researchers, cited or not here, who used their expertise and time to understand and improve the management of the seed conditions covered in the present review. Many thanks!

**Conflicts of Interest:** The authors declare no conflicts of interest.

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
