# Peer review of "Soybean Seed Coat Cracks and Green Seeds—Predisposing Conditions, Identification and Management"

_2674-1024, doi:10.3390/seeds3010011_

Round 1

Reviewer 1 Report

Comments and Suggestions for Authors

This team has authored a review of the literature on two issues that influence soybean quality. The subject is relevant to the journal and the writing style for sentence structure is good. There seems to be repetition and a revision could probably reduce the length substantially without losing content. I am not an agronomist so I am not familiar with the soybean literature. My feeling is that the authors have not done an adequate job justifying the review. Also there are unpublished data in the review, and this is disallowed by most journals.

A good review will begin by citing other similar reviews then explaining why the new review is needed. In other words, what is being added to what has already been reviewed. The reader should not be expected to do that. The authors did not even attempt this in the introduction. One review is referenced in lines 228-232. Why is this buried so deep in the manuscript? This needs to be moved to the Introduction and other soybean quality reviews added. Even if cited reviews are several years old, they still need to be noted so the reader understands the need for a new review. A clear statement about why this new review is needed must follow these citations.

Section 6 is very pragmatic and provides actionable science. The other sections are more ambiguous. More of the manuscript needs to be like Section 6.

Lines 12 and 40. The word serosity is not a word. It needs to be deleted or defined in the text.

I believe there is so much written about soybean that there is no justification to cite work on other legumes or other crops. Surely all of the points that were made with other species could be made with soybean literature. Text addressing other crops begin on:

Lines 164, 170, and 412-458.

Line 165. Careful should be care.

Line 167. Calibrate should be calibrated.

Lines 198-202, lines 269-273, and lines 517-520 provide a list of factors that influence the seed traits, but the manuscript never develops this. There should be a section added that explains how these factors influence seed quality, not just a list of the factors. The section should be written in the style of Section 6.

Lines 216-221. Please break this into two sentences.

Line 263. I don’t understand how a seed can have humidity? A seed is not a gas. Is this relative water content?

Lines 268-269. This is a profound statement that is buried deep in the manuscript. This should be in the abstract and explained in more detail.

Lines 275-325. This does not belong in a bona fide review. It is newly published information. Including this means the manuscript is not a review.

Lines 381-383. Please rewrite.

Section 7 is far too short. It is also rather weak. With the immense literature on soybean surely the authors can put much more detail into what needs to be done.

Author Response

Thanks for your time and contributions to improving the review “Soybean seed coat cracks and green seeds - predisposing conditions, identification and management”.

Most of the suggestions and corrections cited were implemented according to your comments; however, some points need an answer due to the non-implementation of such points.

  1. You mentioned in “Comments and Suggestions for Authors
” that “A good review will begin by citing other similar reviews then explaining why the new review is needed…”.

We extensively researched the literature for the latest reviews on seed crack and greenish, focusing on soybeans. Still, little to nothing was found, and most findings were reports of or some dimensioning of these seed stresses, so we tried to assemble as many pieces of information as possible in this updated review. A few reviews related to these seed issues were included where they should contribute more to the context in which they were included.

To our knowledge, this is one of the first reviews on such exclusive issues (seed coat cracks and green seeds) on soybeans, and it will be very cited. This review includes reports of the same seed issues occurring for other plant species due to the scarcity-specific literature, but it mainly presents soybeans. Additionally, this review highlights strategies that can be used to reduce the occurrence and severity of such seed stresses.

  1. You mentioned in “Comments and Suggestions for Authors
” that “Section 6 is very pragmatic and provides actionable science. The other sections are more ambiguous. More of the manuscript needs to be like Section 6.”.

Section 6 deals with the “crop desiccation as a crop management tool” for uniform harvests. This topic has many reports in the literature, not only for soybeans but for many other crops; thus, more results are available to build up Section 6. Additionally, the intention of Section 6 was to provide much information about the use of herbicides (active ingredient) with their respective doses and results showing that many aspects of the environment (soil, climate) and plants must be considered before crop desiccation. Other crop management, such as crop fertilization, depends on fewer variables, such as soil analyses and the historical fertilizer uses in the area, with no wide range of possible results coming out. However, changes were made throughout the manuscript to improve text information and fix ambiguities.

  1. You mentioned in “Comments and Suggestions for Authors
” that “I believe there is so much written about soybean that there is no justification to cite work on other legumes or other crops…”.

Not many articles have been published about those soybean seed issues (seed coat cracks and green seeds), and most were cited in our review. That is why we also reported these seed issues on other crops, so the experience with other crops could be used as a reference to approach soybean management.

Other suggestions you made were all considered and implemented as well as we could. We hope we have reached your expectations and that the manuscript can be published soon. If needed, another round of corrections and improvements will be carefully read by the authors and implemented until the manuscript reaches the standard for publication in Seeds journal.

Reviewer 2 Report

Comments and Suggestions for Authors

The author clearly defines the scope and purpose of the review paper. I really enjoy reading this article. However, I believe authors must consider these suggestions for improving the quality of the paper.

Firstly, I noticed that the discussion on Raman spectroscopy appears unrelated to the main topic. I recommend excluding this section to maintain focus.

Additionally, I recommend exploring Soybase for comprehensive genetic studies on seed coat cracks and seed color. Numerous studies have identified numerous QTLs and candidate genes for these two traits. To enhance the paper, consider adding two dedicated sections: 1) Seed Coat Cracks and 2) Seed Color. In these sections, summarize and categorize all discovered QTLs and genes, providing insights into the types of studies conducted so far. Conclude by highlighting any identified research gaps in these traits. This approach will enhance the paper's depth and relevance.

Comments on the Quality of English Language

In total was good.

Author Response

Thanks for your time and contributions to improving the review “Soybean seed coat cracks and green seeds - predisposing conditions, identification and management”.

All the suggestions and corrections cited were implemented according to your comments. We hope we have reached your expectations and that the manuscript can be published soon.

If needed, another round of corrections and improvements will be carefully read by the authors and implemented until the manuscript reaches the standard for publication in Seeds journal.

Reviewer 3 Report

Comments and Suggestions for Authors

Although this manuscript is a general review and not a systematic literature review, it is still suggested that the author supplement the time frame and screening criteria for literature search.This manuscript mainly focuses on the review of the seed coating technology , it is suggested to increase the  discussion on mechanism of seed coating.

Introduction is too long, and it is suggested to keep it concise and to the point. Please rework the introduction.

Authors should provide some suitable references to support seed coating.

References are not as per author guideline of the Seeds

Author Response

Thanks for your time and suggestions for improving the review “Soybean seed coat cracks and green seeds - predisposing conditions, identification and management”. However, a point needs to be clarified.

You mentioned in “Comments and Suggestions for Authors
” that “This manuscript mainly focuses on the review of the seed coating technology; it is suggested to increase the discussion on the mechanism of seed coating”.

This review is not about seed coating technology [application of exogenous materials onto seed surface to improve seed characteristics and/or deliver active compounds (e.g., plant growth regulators, micronutrients, microbial inoculants) that can protect the seed].

The seed coat considered in this review is the outermost natural layer (cuticle + epidermis + hypodermis + parenchyma) of the seed, originating from the integuments of the ovule. Such an organ (seed coat) is subject to conditions during seed development that can generate ruptures to its integrity and greening of the seed. Both seed issues reduce seed quality and viability.

We hope we have reached your expectations and that the manuscript can be published soon. If needed, another round of corrections and improvements will be carefully read by the authors and implemented until the manuscript reaches the standard for publication in Seeds journal.

Round 2

Reviewer 1 Report

Comments and Suggestions for Authors

The authors have done an adequate job in the revision. 

Author Response

Dear reviewer 1, thanks!

Reviewer 3 Report

Comments and Suggestions for Authors

Dear authors,

Your manuscript still needs some corrections as follows:

1)      One review is referenced in lines 234-238.

2)      Lines 200-207, lines 274-282, and lines 577-581 provide a list of factors that influence the seed traits, but the manuscript never develops this. There should be a section added that explains how these factors influence seed quality, not just a list of the factors.

3)      Lines 275-282. This is a profound statement that is buried deep in the manuscript. This should be in the abstract and explained in more detail.

4)      Section 7 is far too short. It is also rather weak. With the immense literature on soybean surely the authors can put much more detail into what needs to be done.

5)      References are not as per author guideline of the Seeds. Author 1, A.B.; Author 2, C.D. Title of the article. Abbreviated Journal Name Year, Volume, page range.

6)      This manuscript mainly focuses on the review of the seed coat cracks and green seeds in soybean, it is suggested to increase the discussion on this subject.

7)      There is not citation in introduction. Authors need to provide some suitable references to support here. Introduction is too short and should be the review's main part.

please provide detailed responses to all review reports as well, and highlight all changes in your manuscript.This will facilitate the peer review process.
